# Can Fitness Education Programs Satisfy Fitness Professionals’ Competencies? Integrating Traditional and Revised Importance-Performance Analysis and Three-Factor Theory

**DOI:** 10.3390/ijerph17114011

**Published:** 2020-06-04

**Authors:** Gordon Chih Ming Ku, Chi-Ming Hsieh

**Affiliations:** 1Department of Social Sport, Lingnan Normal University, No.29, Cunjin Rd., Chikan Dist., Zhangjiang City 524048, Guangdong, China; 2International Bachelor Program of Agribusiness, National Chung Hsing University, No.145, Xingda Rd., South Dist., Taichung City 402, Taiwan; hsiehch9@gms.ndhu.edu.tw

**Keywords:** fitness club, competency, evaluation, three-factor theory, revised importance-performance analysis

## Abstract

The purpose of this study is to assess whether fitness education programs could meet the needs or competencies of fitness professionals such as personal trainers and group fitness instructors. A mixed method was adopted to address the objectives of the study. In the first step, a semi-structured interview was conducted with five fitness experts to identify the five dimensions of professional competencies. In the second step, an online survey and paper questionnaires were utilized to collect data from 324 eligible subjects. Traditional importance-performance analysis, revised importance-performance analysis, and the three-factor theory were used to analyze the collected data. The results indicate that “professional skill,” “career development,” and “public relations” are the three most critical professional competencies. “Nutrition” and “coping with stress” should be strengthened and improved in fitness education programs. “Administrative management” is the least important professional competency. Multi-competencies development and lifelong learning are the factors for a successful fitness trainer.

## 1. Introduction

Sports clubs are rapidly growing globally because regular and appropriate physical activity has a profoundly positive effect on health and well-being [1]. With increasing market demand and competition, the need for fitness professionals is also steadily increasing, and their professional competencies are therefore certain to carry extra weight. Fitness professionals play a key role in delivering accurate knowledge and skills to clients and motivating them to get involved in fitness activities [2]. Puente and Anshel [3] found that the instructor’s perceived interaction style positively influences the exerciser’s emotional and behavioral consequences of physical activity and increases their motivation to exercise. Accordingly, a competent fitness professional is regarded as an essential resource in the delivery of fitness knowledge and skills for increasing physical activity [4].

A qualified fitness professional has to complete a series of professional education courses and pass rigorous qualification exams which ensure that the fitness trainer has the sufficient competencies to safely and correctly guide clients in fitness activity and also certifies that fitness professionals have the knowledge and skills to assess, motivate, and train clients in their health and fitness needs [5,6]. Zenko and Ekkekakis [7] opined that fitness trainers and instructors should obtain professional certifications from fitness education programs and keep themselves updated periodically. Therefore, professional education programs could determine if fitness professionals possess sufficient knowledge and skills to meet their clients’ fitness needs.

Fitness professionals require extensive knowledge to take responsibility for various training behaviors and health activities because their roles are both complex and multidimensional [8,9]. The American College of Sports Medicine (ACSM) [10] proposes that a competent fitness trainer should possess the competencies of partnerships, data and scientific information, planning and evaluating, interventions, organizational structure, and exercise science in a public health setting and can be categorized into three core competencies: knowledge, skill, and ability [11]. In another study conducted by Malek et al. [6], the researchers developed a scale and fitness instructor knowledge assessment to assess the fitness trainers’ professional competencies, including “nutrition, health screening, testing protocols, exercise prescription, and general training regarding special populations.” Therefore, fitness trainers’ professional competencies include not only fitness training but also other knowledge and skills, such as social interaction and people–space management. However, these indirect competencies are excluded or seldom included in professional fitness education curricula.

Fitness professionals can be categorized into personal trainers (PT) and group fitness instructors (GFI) according to the fitness course’s aims. PTs “develop and implement personalized exercise programs for individuals across a diverse set of health and fitness backgrounds, from professional athletes to individuals only recently cleared to exercise” [12]. GFIs “teach, lead, and motivate individuals through intentionally-designed exercise classes… not only do they excel at planning effective, exercise science-based group sessions for different fitness levels, they also possess a wealth of motivational and leadership techniques that help their classes achieve fitness goals” [13]. The preparation, instruction, and assessment in fitness courses are significantly different for PTs and GFIs. Chiu, Lee, and Lin [14] found four core competencies of PTs using semi-structured interviews: the ability in (1) theory and practice, (2) course planning and design, (3) management and marketing, and (4) facility safety maintenance and healthcare. Change and Kim (8) identified seven critical competencies for fitness club instructors: (1) personal traits, (2) professionalism, (3) fitness and sporting abilities (4) management skills, (5) education and experiences, (6) instructional skills, and (7) service orientation. Thus, an understanding of PTs’ and GFIs’ professional competencies needed in fitness clubs is necessary for clients’ perceived service quality and fitness professionals’ career tenure. Unfortunately, few studies have explored the professional competencies in practice and satisfaction of fitness education programs for PTs and GFIs. Although Jankauskiene [15] conducted a study that tested the professional competencies of health and fitness instructors, the findings only revealed the pass rate of the core and specific knowledge test for fitness and group fitness instructor. The study lacks insight into the correlation between the essential professional competencies and the fitness education programs. This study evaluates fitness education programs and determines the essential competencies for PTs and GFIs. Accordingly, the objectives of this study are: (1) to understand whether fitness education programs can satisfy fitness professionals’ competency needs in fitness clubs, and (2) to identify the similarities and dissimilarities of professional competencies for PTs and GFIs.

Importance-performance analysis (IPA) has been commonly used in various academic fields because it is a useful approach in assessing customer satisfaction and management strategies [16]. It offers useful information by using the dimensions of importance and performance on a perceptual map divided into four quadrants: keep up the good work, concentrate here, low priority, and possible overkill [17]. It reveals what product or service attributes need to be improved immediately, retained, ignored, or shifted to another area (Figure 1) [17,18].

The four quadrants indicate the different meanings for important competencies and professional education programs. In the keep up the good work (high performance–high importance) quadrant, the professional education program satisfies the fitness professionals’ competency needs and the advantage should be retained; the concentrate here (high importance–low performance) quadrant indicates that the professional education program cannot meet the competency needs of fitness professionals and requires immediate improvement; the low priority (low performance–low importance) quadrant indicates that the professional education program and fitness competencies are unimportant and can be ignored temporarily; and, finally, the possible overkill (low importance–high performance) quadrant indicates that professional education programs are not important to fitness professionals and involve too many resources that should be shifted to areas that require improvement.

IPA has been used to analyze the fitness club management. A number of studies applied IPA to explore the fitness club’s service quality [19,20,21,22], fitness program attributes [23], information systems [24], customer value [25], customer quality requirement [26], and priority elements of management [27]. The findings provide insight into improving fitness club management. However, studies into the issue of fitness professionals’ competencies have not adopted IPA to analyze the importance of the practical application and the performance of the fitness education program. This can offer useful information to increase the quality of education programs and enhance the fitness professionals’ competencies.

Matzler, Bailom, Hinterhuber, Renzl, and Pichler [28] have proposed that two relevant implicit hypotheses must be confirmed, or the interpretation of the results will have traditional IPA (TIPA) statistical bias. First, the dimensions of importance and performance should be independent. Second, the performance attributes and overall satisfaction should be linear and symmetrical [28]. Numerous studies, however, had rejected the two implicit hypotheses. For example, Matzler et al. [28] found that the dimensions of importance and performance indicate a causal relationship (importance depends on performance). Furthermore, Matzler et al. [28] and Matzler, Sauerweinb, and Heischmidtc [29] demonstrated that the relationship between performance attributes and overall performance is not linear and asymmetrical. Hence, the failure of the hypotheses could prove to be worthless and false on the TIPA results. To deal with TIPA, therefore, Deng [30] proposed a revised approach with four steps to solve the shortcomings. First, the attributes’ performance (AP) is transformed into a natural logarithm to capture diminishing returns or sensitivity for independent variables in the partial correlation model. The equation is shown below:AP → In (AP_i_) i = 1, 2, 3, …, n (n = total number of attributes)(1)

Second, each natural logarithm of AP is regarded as a variable; the others are fixed with overall performance satisfaction and computed into a multivariate correlation model. Third, partial correlation analysis is used to calculate the implicitly derived importance of the attributes’ absolute coefficient. Finally, the AP and the implicitly derived importance of the absolute coefficients are utilized to divide the perceptual map into four quadrants. Deng [30] proposed that multicollinearity could be avoided with partial correlation. The revised IPA (RIPA) can reduce the bias gap between importance and performance.

The three-factor theory is another assessment approach in customer satisfaction [31] (Figure 2). It consists of three dimensions: basic factors, excitement factors, and performance factors, which are used to explain three different conditions of customer satisfaction. Specifically, the professional education program satisfaction and important competencies are served as the axes; the basic factors denote that fitness professionals are dissatisfied with the professional education program as the program does not meet their professional competency needs. However, when the program meets their professional competency needs, it may not lead to satisfaction for fitness professionals. The fulfillment of basic needs is a necessary but not a sufficient condition for satisfaction [32]. The excitement factors denote that the fitness professionals would be satisfied if the professional education program meets their professional needs, but would not cause dissatisfaction if their competency needs are not delivered in the program. A positive performance on the professional education program has a more significant impact on overall satisfaction than negative performance [32,33]. Finally, the performance factors denote that the fitness professionals feel satisfied if the education program fulfills or exceeds their expectations and feel dissatisfied if their needs are not fulfilled by the program [32,33].

The three-factor theory can be separated into four quadrants of basic, excitement, high performance, and low performance using the explicit importance of TIPA and implicit RIPA [32,34]. The basic quadrant indicates the attributes with low implicit and high explicit importance. That is, fitness professionals claim some competencies as important, but these do not affect satisfaction if their needs are met or exceeded by the program. These professional competencies can be considered the basic competencies to become a qualified fitness professional. Conversely, the excitement quadrant indicates the attributes with high implicit and low explicit importance. Specifically, fitness professionals regard these competencies as unimportant and they do not cause dissatisfaction if the program fulfills or does not fulfill their needs. These professional competencies are regarded as extra capabilities for fitness professionals in fitness clubs. High- and low-performance quadrants indicate that fitness professionals’ important competencies can be differentiated into high- and low-importance performance using the score level of importance [32,34]. To rigorously assess fitness competencies, Liu [32] has suggested that combining multiple assessment approaches to exploring customers’ satisfaction could offer insights into their attitudes towards the products or services. Kuo, Chen, and Deng [35] also proposed that the combination of TIPA, RIPA, and three-factor theory could compensate for the three-factor theory’s limitation of ignoring attribute performance and importance and TIPA’s limitation of considering only dimensional qualities, and enable managers to grasp accurate user perceptions of quality attributes and corresponding coping strategies. TIPA is still used as one of the assessment approaches because it has certain reference value for practical application. This study identifies the essential competencies using three assessment approaches, TIPA, RIPA, and the three-factor theory, simultaneously.

## 2. Methodology

### 2.1. Participants

A purposive sampling technique was employed to recruit eligible respondents. The purposive sampling technique enables the researcher to select suitable informants. They are knowledgeable about the research topic so that it would be most beneficial to get answers to the research questions and meet the research objectives [36]. The eligible PTs had: (a) PT license (s) and experience in instructing individual fitness activities in clubs. On the other hand, the eligible GFIs had: (a) GFI license (s) and experience in instructing group fitness activities. Both PTs and GFIs were full-time employees in the fitness club; the part-time fitness trainers and instructors were excluded. Four hundred questionnaires were distributed among 62 fitness clubs (two large fitness club chains) in Taiwan, and a total of 324 valid responses were obtained (184 from PTs and 140 from GFIs), yielding a response rate of 81.0%.

### 2.2. Data Collection

A mixed method was designed to explore the satisfaction with the professional education program and the important competencies for PTs and GFIs. Data collection was completed in two stages. In the first stage, the researchers invited three scholars teaching in the field of physical fitness, and leisure and sport at different Taiwanese universities, and two fitness experts, who have been in senior manager positions in fitness clubs for over 20 years, to join a semi-structured interview. Three open-ended questions were used to elicit the PTs’ and GFIs’ professional competencies from the interviewees: “What professional competencies should a PT and GFI have in the clubs?”, “Do you think any part of the education programs need improvement for PTs and GFIs?”, and “Do you have any suggestions for PTs’ and GFIs’ careers?” Probing skill was used to elicit information from the interviewees to obtain rich data. The collected data was analyzed using content analysis. A total of 41 professional competencies of PTs and GFIs were identified by open coding, and five dimensions were identified through axial coding: “professional knowledge,” “professional skill,” “public relations,” administrative management,” and “career development.” A structured questionnaire was developed using these fitness competencies and dimensions. Content validation was carried out to ensure reliability and validity. Anastasi [37] stated that content validity could be considered construct validity, and an “expert” agreement can be used to assess the domain and facets of the instrument [38]. Accordingly, one professor and one expert with rich experience in research/theory and practical aspects of fitness clubs were invited to evaluate the structured questionnaire’s contents. Ambiguous sentences, unappropriated terms, and the framework of scale were modified following two experts’ comments to achieve acceptable reliability and validity.

In the second stage, the structured questionnaire was further revised according to the experts’ comments. Thus, an additional item, overall education program satisfaction, was added to the questionnaire to facilitate the drawing of a perceptual map with the RIPA approach [30]. A total of 42 items were used in the final survey, including 41 professional competencies and one overall education program satisfaction. A 5-point Likert-type scale was adopted for the measurement of essential competencies and program satisfaction, with the descriptive equivalents ranging from “strongly dissatisfied” = 1 to “strongly satisfied” = 5 and “strongly unimportant” = 1 to “strongly important” = 5, respectively.

### 2.3. Data Analysis

The collected data were analyzed using descriptive statistics (mean, SDs, and percentage) for the demographic (frequency and percentage), important competency scores (mean and SDs), and program satisfaction scores (mean and SDs). Furthermore, natural logarithm and partial correlation analysis were used to calculate the absolute coefficient of the implicitly important competency instead of the important competency mean scores of TIPA for conducting RIPA and three-factor model. Finally, the mean scores of the program satisfaction and the important competency were applied to draw TIPA perceptual map. The mean scores of the program satisfaction and the implicitly important competency coefficient were further utilized to construct RIPA perceptual map. To establish three-factor mode, this study used the implicitly important competency coefficient and the mean scores of the important competency.

## 3. Results

### 3.1. Demographic Profile of the Respondents

The demographic characteristics of fitness professionals are summarized as follows. Slightly over half of the respondents were male (51.9%). The majority of the sample (91.2%) had an undergraduate degree. The largest age group was 21–30 years old (75.7%). About 53.3% of the fitness professionals had graduated from departments of physical education or physical leisure management. Most of the respondents had worked in the fitness industry for more than one year (69.7%). One in three fitness professionals possessed a national fitness license (37.3%), and 29.9% of the fitness professionals had an international fitness license, such as Aerobics and Fitness Association of America (AFAA), American College of Sports Medicine (ACSM), or National Strength and Conditioning Association (NSCA), etc.

### 3.2. Perceptions of Education Program Satisfaction and Importance of Professional Competencies

According to the reliability test, the PTs’ Cronbach’s alpha of the five dimensions (professional knowledge, professional skill, public relations, administrative management, and career development) ranged from 0.94 to 0.98 on education program satisfaction. It ranged from 0.88 to 0.96 on the important competencies (Table 1). Conversely, the GFIs’ Cronbach’s alpha on education program satisfaction and important competencies ranged from 0.81 to 0.95 and 0.92 to 0.98, respectively (Table 2). The reliability of the fitness competency scale was acceptable for both PTs and GFIs.

In the PT sample, 63.8% of fitness competencies (28 competencies) are perceived as highly satisfactory on education programs (M > 4.00), including 11 competencies in professional skills, 11 in professional knowledge, four in career development competencies, and two in public relations competencies (Table 1). Conversely, around 95.1% of the competencies are regarded as highly important competencies (M > 4.00). The top five important competencies are: “revising incorrect exercise execution/technique” (M = 4.77), “identifying incorrect exercise execution/technique” (M = 4.75), “exercise execution skill” (M = 4.74), “enthusiasm for job” (M = 4.74), and “fitness assessment skill” (M = 4.73). Furthermore, the coefficients of the implicitly important competencies range from 0.20 to 0.01. “Foreign language” (R = 0.20), “exercise execution skill” (R = 0.18), “life-long learning” (R = 0.17), “design exercise prescription” (R = 0.15), and “financial management” (R = 0.15) are the top five implicitly important competencies.

In the GFI samples, 53.7% of the fitness competencies (22 competencies) were regarded as highly satisfactory on education programs (mean scores over 4.00), including 11 competencies in professional skills, 6 in professional knowledge, 4 in career development, and 1 in public relations (Table 2). Conversely, 82.9% of the competencies were strongly explicitly important (M > 4.00). The top five important competencies were “instructing skill” (M = 4.34), “instructing strategy” (M = 4.33), “instructing demonstration” (M = 4.33), “instructing operation” (M = 4.33), and “positive attitude” (M = 4.33). Moreover, the coefficients of the implicitly important competencies range from 0.28 to 0.01. “Coping with stress” (R = 0.28), “exercise execution skill” (R = 0.23), “social psychology” (R = 0.21), “positive attitude” (R = 0.18), and “instructing operation” (R = 0.17) are the top five implicitly important competencies.

Hence, the GFI samples (M = 4.35) and PT samples (M = 4.15) had a high level of overall satisfaction with the education programs. However, the PT samples perceived a higher level of explicit importance on the dimensions of professional knowledge (PT sample = 4.04 to 4.66; GFI sample = 3.72 to 4.16), public relations (PT samples = 4.21 to 4.54; GFI samples = 3.73 to 4.01), and administrative management (PT samples = 3.90 to 4.23; GFI samples = 3.66 to 3.98) than the GFI samples. The top five explicit and implicitly important competencies in the PT and GFI samples were significantly different. This discrepancy implies that education programs should be fully differentiated according to their competency needs.

### 3.3. TIPA, RIPA, and the Three-Factor Theory Quadrants of PTs and GFIs

The TIPA, RIPA, and the three-factor theory were subsequently conducted, and six perceptual maps were created for the PT and GFI samples, respectively (Figure 3 and Figure 4). The results indicated that there were several differences in the perceptions of program satisfaction and important competencies between PTs and GFIs. For example, human anatomy, exercise psychology, kinesiology, social communication, social interaction, membership recruitment, and promotional activities. Conversely, there were similar viewpoints on the dimensions of professional skill and career development in PTs and GFIs. The following sections present the distribution of attributes on the perceptual maps of the TIPA, RIPA, and the three-factor the theory.

#### 3.3.1. Professional Knowledge

There were considerable differences in the perceptions of the TIPA perceptual map of the PTs and GFIs. Ten competencies, 83.3%, in professional knowledge, were in the “keep up the good work” quadrant in the PTs’ perceptual map: human anatomy, exercise physiology, exercise psychology, kinesiology, emergency medical knowledge, exercise injuries, physical fitness, exercise and health, treatment of exercise injuries, and fitness consultation. Conversely, merely five competencies, 41.7%, in professional knowledge, were regarded as “keep up the good work” from the GFI perspective: exercise injuries, physical fitness, exercise and health, treatment of exercise injuries, and fitness consultation. Surprisingly, knowledge of human anatomy, exercise physiology, kinesiology, and pathology were considered “low priority” in the GFI viewpoint. Furthermore, nutrition was identified as “concentrate here” by both PTs and GFIs. Accordingly, the importance of professional knowledge application was very different between these two groups.

On the RIPA and three-factor theory perceptual maps, nine professional knowledge (75.0%) competencies were in the quadrants of “concentrate here” (RIPA) and “basic factor” (three-factor theory) from the PT perspective: human anatomy, exercise physiology, exercise psychology, emergency medical knowledge, exercise injuries, physical fitness, exercise and health, treatment of exercise injuries, and fitness consultation. Moreover, six professional knowledge competencies were identified as “concentrate here” and “basic factor” in the GFIs’ perception: emergency medical knowledge, exercise injuries, physical fitness, exercise and health, treatment of exercise injuries, and fitness consultation. These findings imply that most of the competencies that were very important to PTs and GFIs can be considered the basic competencies for a qualified fitness professional. Only “pathology” was consistently agreed on by both PTs and GFIs as belonging in the low priority quadrant.

The PTs and GFIs had conflicting opinions on knowledge of human anatomy, exercise psychology, and kinesiology on the RIPA and three-factor theory results. In the PT viewpoint, these three skills should be satisfied with the education programs in terms of their fitness instruction. Conversely, these three skills were not important and are unsatisfactory on education programs to GFIs.

#### 3.3.2. Professional Skills

Both the PTs and GFIs agreed on professional skills, which are the critical competencies in fitness instruction. In the TIPA results, all professional skills were in the “keep up the good work” quadrant. However, there were some similarities and dissimilarities in the perceptions of the RIPA and three-factor theory perceptual maps between the two groups. Both PTs and GFIs believed that clear instruction, instructing skill, instructing strategy, equipment operating, instructing demonstration, identifying incorrect exercise execution/technique, and revising incorrect exercise execution/technique fall in the quadrants of “concentrate here” (RIPA) and “basic factor” (three-factor theory). These competencies were important to both PTs and GFIs, but the education programs did not satisfy their needs. Additionally, exercise execution skill was identified as “keep up the good work” in the RIPA and “high performance” in the three-factor theory.

In the dissimilarities, design exercise prescription and fitness assessment skills were considered highly important competencies and highly satisfactory on education programs by PTs. However, the GFIs thought that these competencies should be enhanced in the education program. Additionally, the GFIs believed that the instructing operation belongs in the “keep up with good work” quadrant. The PTs thought that that the instructing operation should be improved in the education program on account of its important applications in fitness instruction.

#### 3.3.3. Public Relations

Marketing competency, social psychology, fit with the company, and foreign language were consistently considered “low priority” competencies. Social communication and social interaction were considered “concentrate here” competencies in the PTs’ and GFIs’ perception in TIPA. Conversely, in the PTs’ RIPA perceptual map, all public relations competencies were considered unnecessary. However, the GFIs considered that the public relations competencies of social communication and social interaction fall into the “concentrate here” quadrant. On the three-factor theory perceptual map, only social communication was constantly considered a “basic competency” in the PTs’ and GFIs’ perception. Furthermore, according to the finding of the three-factor thesis analysis, public relations showed significantly diverse competencies in the PTs and GFIs, which verifies that they need different education programs in instructing style and interaction with clients.

#### 3.3.4. Administrative Management

The TIPA results indicated that all administrative management competencies were regarded as unimportant skills from the PT perspective. From the GFI viewpoint, only membership recruitment and promotional activities were considered critical competencies, but they should be enhanced in the education programs. The RIPA results show that all administrative management competencies were in the quadrants of “low priority” and “possible overkill” in the PTs’ and GFIs’ perceptions, which implies that administrative management could be ignored temporarily. Furthermore, most of the administrative management competencies fall into the quadrant of high performance from the PT and GFI perspectives. Only financial management and promotional activities were identified as excitement factor competencies for PTs, which are extra capabilities in fitness clubs.

#### 3.3.5. Career Development

Career development, on the TIPA and RIPA perceptual maps, was unanimously identified as an essential competency for PTs and GFIs. However, coping with stress, positive attitude, and enthusiasm for job should be emphasized more in the education programs, in the PTs’ opinion. The GFIs’ also suggested that enthusiasm for job and life-long learning needs improvement in the education programs. Hence, career development competencies were essential abilities for a sustainable career for both PTs and GFIs and should be given more attention to the education programs. The perceptions of fitness competencies among the PTs and GFIs is shown in Table 3.

The findings indicate significantly dissimilar competencies in fitness instruction for the PTs and GFIs. Merely 19 competencies (46.3%) were consistently agreed on by both. The important competencies in fitness instruction for both PTs’ and GFIs’ included exercise injuries, physical fitness, exercise and health, emergency medical knowledge, treatment of exercise injuries, and fitness consultation (in professional knowledge); clear instruction, instructing skill, instructing strategy, equipment operating, instructing demonstration, identifying incorrect exercise execution/technique, and revising incorrect exercise execution/technique (in professional skills); and enthusiasm for job (in career development). These competencies were considered to be vital abilities of the PTs and GFIs, which should either be retained or improved in the education programs. Conversely, social psychology (in public relations) and administration, membership information management, club business management, space design and management, and sports law (in administrative management) are considered unimportant competencies for the PTs and GFIs. They can be temporarily ignored in the education programs.

## 4. Discussion and Conclusions

This study aims to explore whether the fitness education programs satisfy the fitness professionals’ competencies by integrating TIPA, RIPA, and three-factor theory. This study further explores the similar and dissimilar competencies between them. The findings reveal that professional skill and career development are regarded as vital competencies, but administrative management is considered unnecessary by both PTs and GFIs. Furthermore, the PTs’ and GFIs’ opinions are divided into public relations. Accordingly, fitness education programs for PTs and GFIs should consider balancing the program’s practical needs and content. For example, increasing career development is regarded as an important competency for both PTs and GFIs.

It is important to note that the findings of this study are limited to the conditions in Taiwan. However, the methodology and related concepts of the study can be applied to other countries. Moreover, the study findings merely reflect the opinions of PTs and GFIs regarding the fitness competencies in a specific period. Future studies should consider using a longitudinal approach to comprehensively understand the fitness competency needs for PTs and GFIs. Furthermore, most of the participants in this study are from large fitness club chains. However, PTs’ and GFIs’ need for competencies in smaller fitness club chains or independent fitness clubs might be different and should be further explored to design specific education programs. Moreover, constructing a fitness professionals’ competency scale for PTs and GFIs can be considered for future research in the assessment of fitness professionals’ performances and needs and to explore potential connections with other vital variables.

Professional skill and career development are the two most important competencies for both PTs and GFIs, whereas administrative management is an unnecessary competency. Additionally, social communication and social interaction are regarded as vital competencies for PTs and GFIs. Chiu, Lee, and Lin [39] argued that a fitness instructor possessed not only professional knowledge and skill but also interaction and communication with clients. Puente and Anshel [3] proposed that a great instructor’s interacting style facilitates the fulfillment of the client’s needs for autonomy, competence, and relatedness. However, surprisingly, a part of professional knowledge is considered unnecessary (i.e., human anatomy, exercise physiology, kinesiology) from the GFI perspective. Malek, et al. [6] suggested that professional knowledge, skills, and abilities are essential for an effective fitness instructor, even a GFI. Therefore, professional knowledge should not be ignored in GFIs’ education program because of its implication in the instruction process. 

Administrative management is a neglected competency for both PTs and GFIs and in the professional education program. However, Alexandris, Dimitriadis, and Kasiara [40] argued that creating a social environment is important to build service quality, and this can be achieved by enhancing instructors’ and trainers’ administrative management ability. Furthermore, administrative management helps build service quality standards and creates a standard of self-management for instructors and trainers to improve their professional competencies [14, 39]. Accordingly, the professional education program designer should consider adding a few administrative management courses to increase the PTs’ and GFIs’ serviceability and self-improvement.

Nutrition should be improved in the PTs’ and GFIs’ professional education programs because most of the courses focus on professional knowledge and instructional skills. However, there is a significant relationship between nutrition and fitness training. Barnes, Ball, and Desbrow [41] stated that fitness professionals are well placed to promote healthy dietary behaviors in clients and motivate them to change their lifestyle. In Australia, a qualified personal trainer can provide basic healthy eating advice using the nationally endorsed nutrition guidelines [42], which improves the clients’ physical health and enhance the effect of fitness training. Thus, the program designer should consider basic nutrition knowledge in professional training courses for PTs and GFIs.

Career development is an essential dimension for both PTs and GFIs. Particularly, coping with stress is a core competency for continuing a professional fitness career. Fisher-McAuley, Stanton, Jolton, and Gavin [43] suggested that offering extra training, such as time management, for coping with stress would benefit the fitness trainers’ work commitments. Moreover, enhancing career development curriculum in education program not only increases fitness professionals’ loyalty to their fitness careers but also benefits cost-reduction and service quality for the fitness club. ESSA-Sport [44] mentioned that “better skills supporting a more active society, through better quality services and better business in the sport sector”. Accordingly, the professional education program should consider adding coping strategies to the curriculum on fitness professionals’ work stress, especially for PTs. This study has limitations that need to be addressed, although the study makes some theoretical and managerial contributions to the field of fitness education programs, and fitness professionals’ competencies. The first limitation concerns that the findings only reflected the fitness education program and professional competencies in Taiwan, which may not be appropriate for spreading to other countries. However, the method and the statistic approach can be applied to a similar issue in different countries. Therefore, exploring and comparing the differences between fitness education programs and professional competencies in different countries should be comprehensive in future studies. The second limitation concerns the selection of the participants and adaptation of the constructs or variables (indicators). Future researchers could pay attention to a wider respondent base across other fitness fields, and the selection or development of a well-established scales or measurement items.

## Figures and Tables

**Figure 1 ijerph-17-04011-f001:**
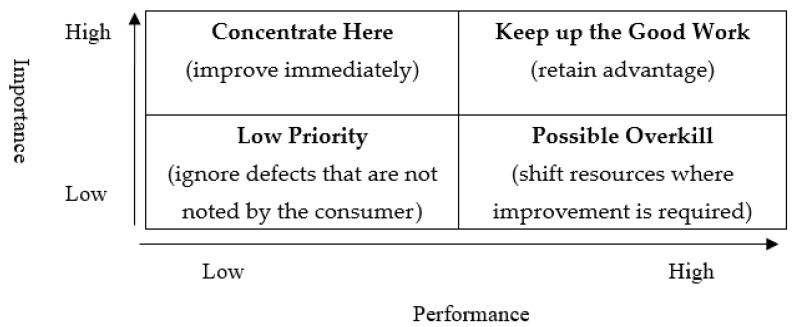
Importance-performance analysis (IPA) perceptual map.

**Figure 2 ijerph-17-04011-f002:**
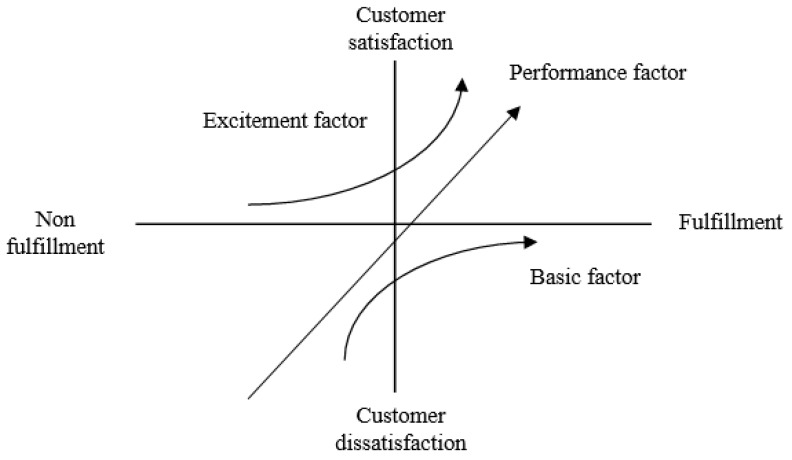
The conceptual model of Three-factor theory.

**Figure 3 ijerph-17-04011-f003:**
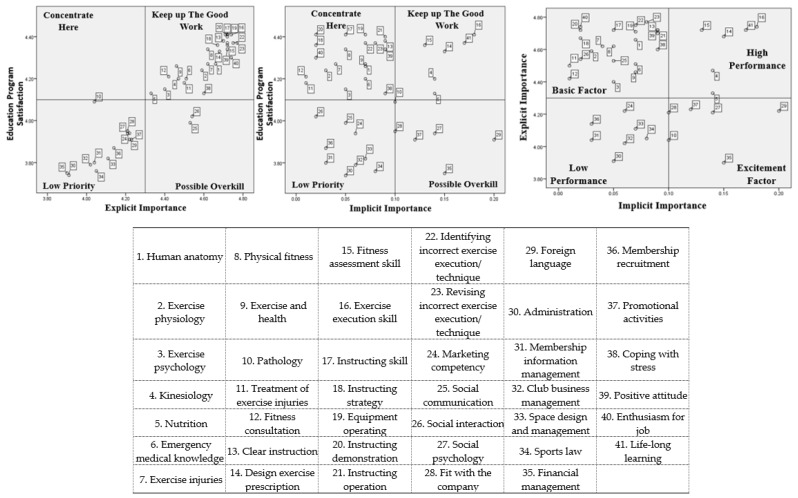
PTs’ perceptual map of the three assessment approaches.

**Figure 4 ijerph-17-04011-f004:**
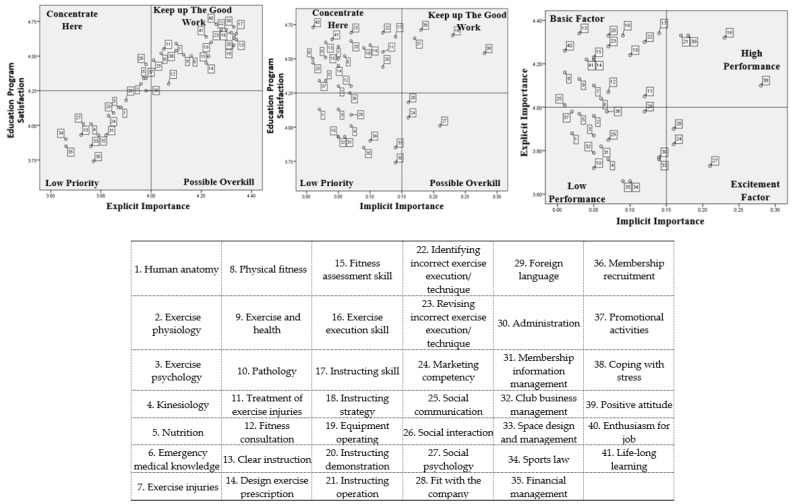
GFIs’ perceptual map of the three assessment approaches.

**Table 1 ijerph-17-04011-t001:** Personal trainers’ (PT) satisfaction, and explicit and implicit importance on competencies.

Dimensions and Items	Satisfaction of Program	Explicit Importance	Implicit Importance
	Mean (SD)	Mean (SD)	Partial Correlation
**Professional knowledge**	**Cronbach’s α = 0.98**	**Cronbach’s α = 0.93**	
1. Human anatomy	4.27 (0.69)	4.66 (0.59)	0.07
2. Exercise physiology	4.24 (0.70)	4.59 (0.55)	0.03
3. Exercise psychology	4.15 (0.77)	4.40 (0.69)	0.05
4. Kinesiology	4.20 (0.74)	4.47 (0.63)	0.14
5. Nutrition	4.13 (0.81)	4.33 (0.68)	0.14
6. Emergency medical knowledge	4.20 (0.77)	4.51 (0.63)	0.07
7. Exercise injuries	4.27 (0.70)	4.62 (0.54)	0.04
8. Physical fitness	4.34 (0.68)	4.62 (0.55)	0.05
9. Exercise and health	4.26 (0.67)	4.46 (0.69)	0.07
10. Pathology	4.09 (0.79)	4.04 (0.81)	0.10
11. Treatment of exercise injuries	4.18 (0.75)	4.50 (0.66)	0.01
12. Fitness consultation	4.21 (0.73)	4.42 (0.69)	0.01
**Professional skill**	**Cronbach’s α = 0.98**	**Cronbach’s α = 0.96**	
13. Clear instruction	4.38 (0.72)	4.70 (0.50)	0.09
14. Design exercise prescription	4.33 (0.72)	4.68 (0.50)	0.15
15. Fitness assessment skill	4.36 (0.71)	4.73 (0.50)	0.13
16. Exercise execution skill	4.41 (0.70)	4.74 (0.46)	0.18
17. Instructing skill	4.41 (0.71)	4.72 (0.46)	0.05
18. Instructing strategy	4.36 (0.70)	4.67 (0.50)	0.02
19. Equipment operating	4.41 (0.69)	4.71 (0.48)	0.07
20. Instructing demonstration	4.41 (0.71)	4.72 (0.47)	0.02
21. Instructing operation	4.40 (0.71)	4.72 (0.50)	0.09
22. Identifying incorrect exercise execution/technique	4.37 (0.74)	4.75 (0.54)	0.07
23. Revising incorrect exercise execution/technique	4.37(0.72)	4.77 (0.47)	0.08
**Public relations**	**Cronbach’s α = 0.97**	**Cronbach’s α = 0.91**	
24. Marketing competency	3.94 (0.84)	4.22 (0.81)	0.06
25. Social communication	3.99 (0.83)	4.53 (0.61)	0.05
26. Social interaction	4.02 (0.85)	4.54 (0.63)	0.02
27. Social psychology	3.94 (0.81)	4.21 (0.76)	0.14
28. Fit with the company	3.95 (0.88)	4.21 (0.78)	0.10
29. Foreign language	3.91 (0.88)	4.22 (0.75)	0.20
**Administrative management**	**Cronbach’s α = 0.98**	**Cronbach’s α = 0.93**	
30. Administration	3.74 (0.90)	3.91 (0.81)	0.05
31. Membership information management	3.80 (0.88)	4.04 (0.77)	0.03
32. Club business management	3.79 (0.89)	4.02 (0.81)	0.06
33. Space design and management	3.82 (0.87)	4.11 (0.78)	0.07
34. Sports law	3.76 (0.92)	4.05 (0.77)	0.08
35. Financial management	3.75 (0.92)	3.90 (0.86)	0.15
36. Membership recruitment	3.87 (0.87)	4.14 (0.84)	0.03
37. Promotional activities	3.91 (0.91)	4.23 (0.78)	0.12
**Career development**	**Cronbach’s α = 0.94**	**Cronbach’s α = 0.88**	
38. Coping with stress	4.13 (0.83)	4.60 (0.57)	0.09
39. Positive attitude	4.34 (0.75)	4.72 (0.53)	0.09
40. Enthusiasm for job	4.30 (0.78)	4.74 (0.53)	0.02
41. Life-long learning	4.37 (0.70)	4.72 (0.55)	0.17

**Table 2 ijerph-17-04011-t002:** Group fitness instructors’ (GFI) satisfaction, and explicit and implicit importance on competencies.

Dimensions and Items	Satisfaction of Program	Explicit Importance	Implicit Importance
	Mean (SD)	Mean (SD)	Partial Correlation
**Professional knowledge**	**Cronbach’s α = 0.89**	**Cronbach’s α = 0.96**	
1. Human anatomy	4.13 (0.76)	3.88 (0.75)	0.02
2. Exercise physiology	4.30 (0.68)	3.96 (0.74)	0.05
3. Exercise psychology	4.13 (0.74)	3.87 (0.80)	0.05
4. Kinesiology	4.01 (0.74)	3.76 (0.77)	0.07
5. Nutrition	4.34 (0.65)	3.97 (0.79)	0.03
6. Emergency medical knowledge	4.52 (0.57)	4.04 (0.77)	0.06
7. Exercise injuries	4.59 (0.52)	4.10 (0.75)	0.05
8. Physical fitness	4.50 (0.58)	4.16 (0.68)	0.01
9. Exercise and health	4.51 (0.54)	4.13 (0.72)	0.03
10. Pathology	3.93 (0.80)	3.72 (0.85)	0.05
11. Treatment of exercise injuries	4.55 (0.54)	4.05 (0.73)	0.12
12. Fitness consultation	4.30 (0.66)	4.07 (0.81)	0.07
**Professional skill**	**Cronbach’s α = 0.95**	**Cronbach’s α = 0.98**	
13. Clear instruction	4.62 (0.57)	4.34 (0.67)	0.03
14. Design exercise prescription	4.45 (0.63)	4.22 (0.73)	0.05
15. Fitness assessment skill	4.50 (0.58)	4.23 (0.72)	0.05
16. Exercise execution skill	4.67 (0.52)	4.32 (0.68)	0.23
17. Instructing skill	4.66 (0.55)	4.34 (0.68)	0.14
18. Instructing strategy	4.58 (0.54)	4.33 (0.72)	0.09
19. Equipment operating	4.60 (0.55)	4.24 (0.73)	0.10
20. Instructing demonstration	4.63 (0.54)	4.33 (0.70)	0.07
21. Instructing operation	4.65 (0.54)	4.33 (0.69)	0.17
22. Identifying incorrect exercise execution/technique	4.69 (0.58)	4.30 (0.70)	0.12
23. Revising incorrect exercise execution/technique	4.69 (0.48)	4.28 (0.73)	0.07
**Public relations**	**Cronbach’s α = 0.81**	**Cronbach’s α = 0.94**	
24. Marketing competency	4.07 (0.73)	3.83 (0.81)	0.16
25. Social communication	4.47 (0.58)	4.01 (0.82)	0.01
26. Social interaction	4.44 (0.60)	3.98 (0.84)	0.12
27. Social psychology	4.01 (0.66)	3.73 (0.85)	0.21
28. Fit with the company	4.18 (0.71)	3.90 (0.86)	0.16
29. Foreign language	4.09 (0.78)	3.85 (0.84)	0.07
**Administrative management**	**Cronbach’s α = 0.91**	**Cronbach’s α = 0.95**	
30. Administration	3.74 (0.74)	3.77 (0.84)	0.14
31. Membership information management	3.93 (0.77)	3.82 (0.79)	0.06
32. Club business management	3.93 (0.76)	3.79 (0.81)	0.05
33. Space design and management	3.85 (0.77)	3.76 (0.80)	0.14
34. Sports law	3.90 (0.84)	3.66 (0.85)	0.10
35. Financial management	3.85 (0.83)	3.66 (0.84)	0.09
36. Membership recruitment	4.25 (0.72)	3.98 (0.73)	0.07
37. Promotional activities	4.34 (0.68)	3.98 (0.80)	0.02
**Career development**	**Cronbach’s α = 0.86**	**Cronbach’s α = 0.92**	
38. Coping with stress	4.54 (0.54)	4.10 (0.83)	0.28
39. Positive attitude	4.71 (0.53)	4.33 (0.70)	0.18
40. Enthusiasm for job	4.73 (0.49)	4.26 (0.75)	0.01
41. Life-long learning	4.64 (0.54)	4.22 (0.75)	0.04

**Table 3 ijerph-17-04011-t003:** The perceptions of fitness competencies among PTs and GFIs.

Fitness Competency	PT			GFI		
Professional Knowledge	Traditional IPA	Revised IPA	Three-Factor Theory	Traditional IPA	Revised IPA	Three-Factor Theory
1. Human anatomy	I	II	BF	III	III	PL
2. Exercise physiology	I	II	BF	II	II	PL
3. Exercise psychology	I	II	BF	III	III	PL
4. Kinesiology	I	I	PL	III	III	PL
5. Nutrition	II	I	PH	II	II	PH
6. Emergency medical knowledge	I	II	BF	II/I	II	BF
7. Exercise injuries	I	II	BF	I	II	BF
8. Physical fitness	I	II	BF	I	II	BF
9. Exercise and health	I	II	BF	I	II	BF
10. Pathology	III	III/IV	PL/EF	III	III	PL
11. Treatment of exercise injuries	I	II	BF	I	II	BF
12. Fitness consultation	I	II	BF	I	II	BF
**Professional skill**						
13. Clear instruction	I	II	BF	I	II	BF
14. Design exercise prescription	I	I	PH	I	II	BF
15. Fitness assessment skill	I	I	PH	I	II	BF
16. Exercise execution skill	I	I	PH	I	I	PH
17. Instructing skill	I	II	BF	I	II	BF
18. Instructing strategy	I	II	BF	I	II	BF
19. Equipment operating	I	II	BF	I	II	BF
20. Instructing demonstration	I	II	BF	I	II	BF
21. Instructing operation	I	II	BF	I	I	PH
22. Identifying incorrect exercise execution/technique	I	II	BF	I	II	BF
23. Revising incorrect exercise execution/technique	I	II	BF	I	II	BF
**Public relations**						
24. Marketing competency	III	III	PL	III	IV	EF
25. Social communication	II	III	BF	II	II	BF
26. Social interaction	II	III	BF	II	II	PL
27. Social psychology	III	IV	EF	III	IV	EF
28. Fit with the company	III	III/IV	PL/EF	III	IV	EF
29. Foreign language	III	IV	EF	III	III	PL
**Administrative management**						
30. Administration	III	III	PL	III	III	PL
31. Membership information management	III	III	PL	III	III	PL
32. Club business management	III	III	PL	III	III	PL
33. Space design and management	III	III	PL	III	III	PL
34. Sports law	III	III	PL	III	III	PL
35. Financial management	III	IV	EF	III	III	PL
36. Membership recruitment	III	III	PL	II	III	PL
37. Promotional activities	III	IV	EF	II	III	PL
**Career development**						
38. Coping with stress	II	II	BF	I	I	PH
39. Positive attitude	I	II	BF	I	I	PH
40. Enthusiasm for job	I	II	BF	I	II	BF
41. Life-long learning	I	I	PH	I	II	EF

Note: I: keep up the good work; II: concentrate here; III: low priority; IV: possible overkill; BF: basic factor; EF: excitement factor; PL: performance low; PH: performance high.

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
