# Peer review of "Can Fitness Education Programs Satisfy Fitness Professionals’ Competencies? Integrating Traditional and Revised Importance-Performance Analysis and Three-Factor Theory"

_ijerph, 2020, doi:10.3390/ijerph17114011_

Round 1
Reviewer 1 Report
Dear Editor, I am very pleased with the revised work.I express a favorable opinion for publication.
The topic is deeply relevant, the analysis is complete and
the background is rich. I hope there will be further developments.
Best regards,
Patrizia Belfiore
Author Response
Dear Editor,
Thank you very much for forwarding to us the valuable comments from the reviewers. Their time and effort are very much appreciated. We have revised our manuscript based on their comments, and have summarized the changes in the following table:
|
Reviewer’s Comments and Changes Made |
Section & Page No. |
|
Reviewer 1 Dear Editor, I am very pleased with the revised work. I express a favourable opinion forpublication. The topic is deeply relevant, the analysis is complete and the background is rich. I hope there will be further developments.
Response and Changes Made Thank you for your support. I hope this paper will be published in the International Journal of Environmental Research and Public Health soon. Thank you!
|
|

Reviewer 2 Report
I thank the Editors for having participated in the review of this study. Likewise, I congratulate the authors for their paper as it adds value and increases knowledge in fitness industry.
However, I have doubts and I contribute different comments to improve the quality of the study.
-There are studies with IPA analysis in the fitness sector, and it is recommended to include a brief introduction.
-It is recommended to review the following study:
Jankauskiene, R. (2018). Professional Competencies of Health and Fitness Instructors: Do they Match the European Standard?. Kinesiology, 50(2), 269-276.
-It is recommended to review the European project:
ESSA Sport – A European Sector Skills Alliance for Sport and Physical Activity
http://eose.org/our_work/essa-sport-a-european-sector-skills-alliance-for-sport-and-physical-activity/
-A greater explanation of the study participants is necessary.
-A greater explanation of the data analysis and validation of the instrument is necessary.
-A further explanation of limitations and future lines of research is necessary.
Author Response
Dear Editor,
Thank you very much for forwarding to us the valuable comments from the reviewers. Their time and effort are very much appreciated. We have revised our manuscript based on their comments, and have summarized the changes in the following table:
|
Reviewer’s Comments and Changes Made |
Section & Page No. |
|
Reviewer 2 I thank the Editors for having participated in the review of this study. Likewise, I congratulate the authors for their paper as it adds value and increases knowledge in fitness industry. However, I have doubts and I contribute different comments to improve the quality of the study. -There are studies with IPA analysis in the fitness sector, and it is recommended to include a brief introduction.
Response and Changes Made Thank you for your comments! We have reviewed and briefly introduced the studies of fitness club with IPA approach in the “Introduction” section (p. 3) as follows: IPA has been used to analyze the fitness club management. The number of studies applied IPA to explore the fitness club’s service quality [19-22], fitness program attributes [23], information systems [24], customer value [25], customer quality requirement [26], and priority elements of management [27]. The findings provide insight into improving fitness club management. However, the issue of fitness professionals’ competencies has not adopted IPA to analyze the importance of practical application and the performance of the fitness education program. That can offer useful information to increase the quality of the education program and enhance the fitness professionals’ competencies.
|
“Introduction” (p.3) |
|
-It is recommended to review the following study: Jankauskiene, R. (2018). Professional Competencies of Health and Fitness Instructors: Do they Match the European Standard?. Kinesiology, 50(2), 269-276.
Response and Changes Made Thank you for your comments! We have reviewed the paper and cited the reference in the “introduction” section (p. 2) as follows: Fitness professionals can be categorized into personal trainers (PT) and group fitness instructors (GFI) according to the fitness course's aims. PTs “develop and implement personalized exercise programs for individuals across a diverse set of health and fitness backgrounds, from professional athletes to individuals only recently cleared to exercise” [12]. GFIs “teach, lead, and motivate individuals through intentionally-designed exercise classes…not only do they excel at planning effective, exercise science-based group sessions for different fitness levels, they also possess a wealth of motivational and leadership techniques that help their classes achieve fitness goals” [13]. The preparation, instruction, and assessment in fitness courses are significantly different for PTs and GFIs. Chiu, Lee, and Lin [14] found four core competencies of PTs using semi-structured interviews: ability in (1) theory and practice, (2) course planning and design, (3) management and marketing, and (4) facility safety maintenance and healthcare. Change and Kim (8) identified seven critical competencies for fitness club instructors: (1) personal traits, (2) professionalism, (3) abilities, (4) management skills, (5) education and experiences, (6) instructional skills, and (7) service orientation. Thus, an understanding of PTs’ and GFIs’ professional competencies needed in fitness clubs is necessary for clients’ perceived service quality and fitness professionals’ career tenure. Unfortunately, few studies have explored the professional competencies in practice and satisfaction of fitness education programs for PTs and GFIs. Although Jankauskiene [15] conducted a study that tests the professional competencies of health and fitness instructors, the findings only revealed the pass rate of the core and specific knowledge test for fitness and group fitness instructor. The study lacks insight into the correlation between the essential professional competencies and the fitness education programs. This study evaluates fitness education programs and determines the essential competencies for PTs and GFIs. Accordingly, the objectives of this study are: (1) to understand if fitness education programs can satisfy fitness professionals’ competency needs in fitness clubs and (2) to identify the similarities and dissimilarities of professional competencies for PTs and GFIs.
|
“Introduction” (p.2) |
|
-It is recommended to review the European project: ESSA Sport – A European Sector Skills Alliance for Sport and Physical Activity http://eose.org/our_work/essa-sport-a-european-sector-skills-alliance-for-sport-and-physical-activity/
Response and Changes Made Thank you for your suggestion! We have reviewed the reference and added it in the “Discussion and Conclusion” section (p. 15) as follows: Career development is an essential dimension for both PTs and GFIs. Particularly, coping with stress is a core competency for continuing a professional fitness career. Fisher-McAuley, Stanton, Jolton, and Gavin [43] suggested that offering extra training, such as time management, for coping with stress would benefit the fitness trainers’ work commitments. Moreover, enhancing career development curriculum in education program not only increases fitness professionals’ loyalty to their fitness careers but also benefits cost-reduction and service quality for the fitness club. ESSA-Sport [44] mentioned that “better skills supporting a more active society, through better quality services and better business in the sport sector”. Accordingly, the professional education program should consider adding coping strategies to the curriculum on fitness professionals’ work stress, especially for PTs.
|
“Discussion and Conclusion” (p.15) |
|
-A greater explanation of the study participants is necessary.
Response and Changes Made Thank you for your comment! We have further explained the participants in the “Methodology” section (p.5) as follows:
A purposive sampling technique was employed to recruit eligible respondents. The purposive sampling technique enables the researcher to select suitable informants. They are knowledgeable about the research topic so that it would be most beneficial to get answers to the research questions and meet the research objectives [36]. The eligible PTs had: (a) PT license(s) and experience in instructing individual fitness activities in clubs. On the other hand, the eligible GFIs had: (a) GFI license(s) and experience in instructing group fitness activities. Both PTs and GFIs were full-time employees in the fitness club; the part-time fitness trainers and instructors were excluded. Four hundred questionnaires were distributed among 62 fitness clubs (two large fitness club chains) in Taiwan, and a total of 324 valid responses were obtained (184 from PTs and 140 from GFIs), a response rate of 81.0%.
|
“Methodology” (p.5) |
|
-A greater explanation of the data analysis and validation of the instrument is necessary.
Response and Changes Made Thank you for your comment! We have further explained the data analysis and reliability and validity of the instrument in the “Methodology” section (pp.5-6) as follows:
A mixed method was designed to explore professional education program satisfaction and the important competencies for PTs and GFIs. Data collection was done in two stages. In the first stage, the researchers invited three scholars, physical fitness and health professors at a Taiwanese university, and two fitness experts, who have been in senior manager positions in fitness clubs for over 20 years, to join a semi-structured interview. Three open-ended questions were used to elicit the PTs’ and GFIs’ professional competencies from the interviewees: “What professional competencies should a PT and GFI have in the clubs?”, “Do you think any part of the education programs need improvement for PTs and GFIs?”, and “Do you have any suggestions for PTs’ and GFIs’ careers?” Probing skill was used to elicit information from the interviewees to obtain rich data. The collected data was analyzed using content analysis. A total of 41 professional competencies of PTs and GFIs were identified by open coding, and five dimensions were identified through axial coding: “professional knowledge,” “professional skill,” “public relations,” administrative management,” and “career development.” A structured questionnaire was developed using these fitness competencies and dimensions. Content validation was carried out to ensure reliability and validity. Anastasi [37] stated that content validity could be considered construct validity, and an “expert” agreement can be used to assess the domain and facets of the instrument [38]. Accordingly, one professor and one expert with rich experience in research/theory and practical aspects of fitness clubs were invited to evaluate the structured questionnaire's contents. Ambiguous sentences, unappropriated terms, and the framework of scale were modified following two experts’ comments to achieve acceptable reliability and validity.
The collected data were analyzed using descriptive statistics (mean, SDs, and percentage) for the demographic (frequency and percentage), important competency scores (mean and SDs), and program satisfaction scores (mean and SDs). Furthermore, natural logarithm and partial correlation analysis were used to calculate the absolute coefficient of the implicitly important competency instead of the important competency mean scores of TIPA for conducting RIPA and three-factor model. Finally, the program satisfaction mean scores and the important competency mean scores were applied to draw TIPA perceptual map. The program satisfaction mean scores and the implicitly important competency coefficient were utilized to construct RIPA perceptual map. The implicitly important competency coefficient and important competency mean scores were used to establish three-factor model.
|
“Methodology” (p.5-6) |
|
-A further explanation of limitations and future lines of research is necessary.
Response and Changes Made Thank you for your suggestion! We have added the study limitation and future studies in the “Discussion and Conclusion” (p.14) as follows:
This study aims to explore whether the fitness education program satisfies the fitness professionals’ competencies by integrating TIPA, RIPA, and three-factor theory. The findings only reflected the fitness education program and professional competencies in Taiwan, which may not be appropriate for spreading to other countries. However, the method and the statistic approach can be applied to a similar issue in different countries. Therefore, exploring and comparing the differences between fitness education programs and professional competencies in different countries should be comprehensive in future studies.
|
“Discussion and Conclusion” (p.14) |

Reviewer 3 Report
Thank you for your submission. This is an interesting study that addresses the importance of training, education, and competency among fitness professionals. Overall, the paper was well written. The methodology and data analyses were strong. I have two suggestions: 1) Check the verb tense for data; and 2) It would be helpful to have Tables 1, 2, 3 on one page instead of split paging.
Author Response
Dear Editor,
Thank you very much for forwarding to us the valuable comments from the reviewers. Their time and effort are very much appreciated. We have revised our manuscript based on their comments, and have summarized the changes in the following table:
|
Reviewer’s Comments and Changes Made |
Section & Page No. |
|
Reviewer 3 Thank you for your submission. This is an interesting study that addresses the importance of training, education, and competency among fitness professionals. Overall, the paper was well written. The methodology and data analyses were strong. I have two suggestions: 1) Check the verb tense for data; and 2) It would be helpful to have Tables 1, 2, 3 on one page instead of split paging.
Response and Changes Made Thank you for your comments! We have checked the grammar of the manuscript, and also made Table 1 (p. 7) and 2 (p. 8) on one page. However, Table 3 (pp.13-14) is difficult to indicate on one page, we have added the Table title on the second page.
|
|
